# A Single Intradermal Injection of Autologous Adipose-Tissue-Derived Stem Cells Rejuvenates Aged Skin and Sharpens Double Eyelids

**DOI:** 10.3390/jpm13071162

**Published:** 2023-07-20

**Authors:** Masamitsu Ichihashi, Masaki Tanaka, Takashi Iizuka, Hiroko Totsuka, Ekuko Tominaga, Yuka Hitomi, Hideya Ando, Takahiro Nishikata, Ken-Ichi Mizutani

**Affiliations:** 1Kobe University, Kobe 657-8501, Japan; 2Arts Ginza Clinic, Tokyo 105-0004, Japan; tana1417@fancy.ocn.ne.jp (M.T.); Iizuka.med@gmail.com (T.I.); chico.v.roco215@gmail.com (H.T.); ekukokue@yahoo.co.jp (E.T.); yumammamja@gmail.com (Y.H.); 3Department of Applied Chemistry and Biotechnology, Okayama University of Science, Okayama 700-0005, Japan; ando@dac.ous.ac.jp; 4Frontier of Innovative Research in Science and Technology, Konan University, Kobe 658-8501, Japan; nisikata@konan-u.ac.jp; 5Laboratory of Stem Cell Biology, Graduate School of Pharmaceutical Science, Kobe Gakuin University, Kobe 650-8586, Japan; mizutani@pharm.kobegakuin.ac.jp

**Keywords:** autologous adipose tissue, mesenchymal stem cells, intradermal injection, skin rejuvenation, eyelids, hair pore

## Abstract

Facial skin aging is the most visible manifestation of aging in the body. In this study, we aimed to rejuvenate aging skin via a one-time intradermal injection of autologous adipose-derived stem cells (ADSCs). Eight patients were enrolled for study. Photographs of patients taken immediately before and 1, 3, 6, and 12 months after ADSC injections were comparatively evaluated for visible skin manifestations. ADSCs were cultured from the abdominal-skin-derived subcutaneous fat tissue, and 1 × 108 cultured ADSCs were injected intradermally into the facial skin. Cultured myoblasts were incubated with the supernatant derived from ADSCs, and the effect was evaluated via glucose consumption and lactic acid production in the medium. Eight cases showed the shallowing and disappearance of wrinkles, including those of the glabella, lower eyelids, crow`s feet, and forehead and nasolabial grooves, a month to several months after treatment. Double eyelids became prominent, and facial pores significantly reduced in size. These effects lasted for over one year. Myoblasts cultured in the presence of an ADSC-derived exosome were activated compared to that of ADSCs cultured without supernatant. The result supports the role of muscle in ADSC skin rejuvenation. The present study first reports that a single intradermal administration of cultured ADSCs rejuvenates aged facial skin over the course of one year. Further, patients exhibited definite double eyelids and pore shrinkage, strongly indicating the active involvement of muscle, which was supported by an in vitro study. Our study also suggested the important role of biological factors delivered from injected stem cells, although the detailed mechanism of rejuvenation effects of ADSC skin injection remains to be clarified.

## 1. Introduction

Facial skin aging is the most visible manifestation of aging in the body, which declines due to physical, physiological, psychological, and social activities. An aged face is characterized by uneven skin color with pigmented freckles, rough skin texture, wrinkles, and sagging of the skin, which often vary based on ethnic origin.

Intradermal injection of small-molecule hyaluronic acid (HA) has been widely used to treat aging facial skin by replenishing the HA component of the dermis. Further, HA is expected to stimulate the collagen production activity of dermal fibroblasts [1]. However, its therapeutic effects are not maintained over time because it has an absorbable micromolecular structure. Autologous cutaneous fat tissue that is extracted surgically was traditionally used for treating facial soft tissue depressions [2]; in addition, it notably improves skin condition and facial structure, resulting in the rejuvenation of the face.

Mesenchymal stem cells, such as umbilical-cord-, bone-marrow-, and adipose-tissue-derived stem cells (ADSCs), are reported to be effective at rejuvenating aged skin in pre-clinical [3] and in vitro experimental studies [4], but a few clinical studies using cultured ADSCs, reported as review articles, suggest the rejuvenation effect in aged human skin [5]. We reported the efficacy of ADSCs in the recovery of motor and sensory function of stroke patients, and reported the notable rejuvenation effect of a single intravenous systemic administration of ADSCs in some patients [6].

ADSCs are available with less invasive procedures compared to bone marrow stem cells and are reported to enhance cutaneous wound healing, and even extra-cellular vesicles derived from ADSC supernatants have been shown to stimulate collagen synthesis and migration of dermal fibroblasts. Supernatants of cultured ADSCs contain many kinds of trophic and growth factors, such as platelet-derived growth factor (PDGF), vascular endothelial growth factor (VEGF), basic fibroblast growth factor (bFGF), transforming growth factor (TGF) b1, b2, hepatocyte growth factor (HGF), keratinocyte growth factor (KGF), and collagen and fibronectin [7] In a pre-clinical study, ADSCs and their supernatants were reported to diminish photo-aged skin wrinkles via increased collagen synthesis of fibroblasts [7]. Further, intradermal injection of ADSCs [8] and nanofat-derived stromal cells combined with platelet-rich fibrin improved the efficacy of facial skin rejuvenation [9].

In the present study, we aimed to evaluate the regenerative effects of autologous ADSCs injected into the dermal layer of facial skin using an injector with nine needles covering an area of about 2 cm^2^ [10]. Photographs that were taken immediately before and 1, 3, 6, and 12 months after a single ADSC injection were comparatively studied to evalute visible skin manifestations by three persons involved in this research (one medical doctor, one nurse, and one photographer). All eight patients showed profound facial skin rejuvenation 1 month to 6 months after a single intradermal injection of ADSCs. A 58-year-old female and a 53-year-old male showed an excellent decrease in the number and depth of wrinkles at 9 and 5 months after injection, respectively, but unfortunately, we could only evaluate the early effects in two cases, since the other patients visited us several months after treatment. All eight patients showed decreased sagging of the cheek with an apparent shallowing of nasolabial folds. Further, six patients exhibited definite double eyelids two weeks to one month after ADSC injection. In order to evaluate the effect of ADSC injection on the eyelids, we used a formula of b/a (b: maximal exposure length between upper and lower eyelids, a: longitudinal diameter of black eye). An excellent decrease in cheek pore size was observed in most cases. We speculate that muscles and fat tissue in the facial skin may be activated by intradermally injected ADSCs and the change to eyelid shape and pore size, in addition to the increase in collagen, elastic fibers, proteoglycan, and hyaluronic acid in the dermis. The precise mechanisms of skin rejuvenation including eyelid and pore size change caused by ADSCs, however, remains to be clarified, since we have no detailed data regarding the effect of ADSCs on muscle and fat tissue of pre-clinical and clinical studies.

To support a possible involvement of muscle tissue, we conducted an in vitro study to show the effect of the exosome of cultured SDCS on the activation of myoblasts, and found that the lactic acid level significantly increased and the glucose level decreased 24 h after ADSC exosome treatment compared to that of a non-treated control, suggesting the effect of ADSCs on muscle activation in vivo.

The present study is the first in the world to indicate that ADSCs injected once intradermally rejuvenate aged skin over a year and induce sharpened double eyelids, possibly by activating muscle cells in addition to increased synthesis of dermal collagen, elastic fibers, and hyaluronic, which may open up a new way of using ADSCs for skin rejuvenation.

## 2. Patients and Methods

### 2.1. Patients and Demography

Between October 2020 and December 2022, 8 patients were enrolled for our ADSC clinical skin rejuvenation study. Eight patients (six females and two males) aged between 32 and 62 years were included in this study (Table 1).

The study was conducted according to the Declaration of Helsinki Principles using a protocol ethically reviewed and approved by the Arts-Ginza Clinic Ethics Committee. Informed consent to participate in this study was obtained from each subject before commencement of the study.

### 2.2. Preparation of Autologous ADSCs

Briefly, ADSCs were prepared from the subcutaneous fat tissue of the abdominal skin of each patient. Patients were treated with a local anesthetic patch and injection in the skin, approximately 10 cm to the side of the umbilicus, and 2~3 rice-sized pieces of subcutaneous fat tissue were surgically obtained from a 0.7 cm incision. The pieces of fat tissue were cut into 15–20 small pieces, placed on a scaffold of nonwoven fabric painted with hydroxy apatite (BioMiraiKobou, Tokyo, Japan) in culture dishes, and cultured at 5% CO_2_ and 37 °C in medium supplemented with 4% autologous serum for 11 to 13 days. They were then trypsinized (0.25% trypsin, BioMiraiKobo, Tokyo, Japan) and reseeded in T75 flasks and further cultured for approximately 3 days in medium containing 2% serum, after which cells were re-trypsinized and cultured in T300 flasks (BM Equipment, Tokyo, Japan), and then trypsinized again and cultured for 3 days in HyperFlasks (Corning Japan, Tokyo, Japan), before the final cell preparation for the treatment. On the day of transplantation, cells were trypsinized and washed four times with saline, then passed through two filters (40 µm and 100 µm), and an average number of 1.0 × 10^8^ (0.8~1.0 × 10^8^) ADSCs were prepared and resuspended in 40 mL saline and divided into four 1 mL syringes containing 0.9 mL each [6]. The stem cell characteristics of the collected cells were confirmed with flow cytometry using CD 73, CD90, and CD105 for positive antibodies, and CD45 for negative antibodies against stem cells, respectively.

### 2.3. Treatment of Aged Skin with ADSCs

After local anesthesia, each patient’s face was treated with intradermal injections of ADSC cells using the injector with 9 needles covering about 2 cm^2^ per injection. Each type of facial skin, including nasolabial folds, Marionette lines, Golgo lines, jaw line, forehead, upper and lower eyelids, external eye parts, and neck skin, was injected. Further, a one-milliliter syringe with a thirty-gauge needle was used to treat skin areas with strong unevenness. After the injection, skin was cleaned and treated with Gentamicin sulfate ointment to prevent bacterial infection.

### 2.4. Evaluation of Skin Rejuvenation

(1)Visible manifestations of wrinkles in photographs of the forehead, glabella, lower eyelids, crow’s feet, Marionette line, nasolabial groove, and sagging of the lower cheek were evaluated using a VAS (visual analogue scale), with a 5-scale score (0: no, 1: slight, 2: mild, 3: moderate, and 4: severe wrinkle), immediately before and 1, 3, and 6 months after, except in cases 1 and 2, who were evaluated a week after injection, after a single ADSC injection. Scores were comparatively evaluated by three persons involved in this research (one medical doctor, one nurse, and one photographer).(2)The effect on facial hair follicles was evaluated using photographs taken before and 5 or 6 months after ADSC skin injection.(3)The effect on eyelids was evaluated using two methods, one being visual facial photographs and the other a formula of eye exposure ratio b/a: maximal exposure length between the upper and lower eyelid (b) divided by the longitudinal eyeball diameter (a) Further, to compensate for differences in magnification of each photograph, we used a formula of h/a to evaluate the eyelid ratio distance (h: eyelid distance of the upper most and lower most of opened eye, a: longitudinal diameter of black eye).(4)The effect of the ADSC supernatant on cultured myoblast cell activity was evaluated.

To support a possible rejuvenation effect of the ADSC injection on muscle tissue, an in vitro study using cultured mouse myoblasts (C2C12 cell) was conducted to study the effect of a cultured ADSC-derived exosome on muscle cell activity by evaluating glucose consumption and lactic acid production. In brief, ADSCs were cultured with DMEM plus 2% sodium pyruvate for 48 h, and the medium was collected and used as an exosome fraction. C2C12 cells seeded in tissue with 12 wells were cultured with high-glucose DMEM via the addition of 1% fetal bovine serum and 50% ADSC condition medium for 24 h, 48 h, or 72 h before sample collection for biological assay.

## 3. Results

(1)Effect of a one-time injection of ADSCs on wrinkle and sagging

The wrinkle scale of five female patients (cases 1~5, Figure 1) was evaluated before and one month after ADSC treatment, and in case 6 (Figure 2), it was evaluated before and 9 months after treatment. The scores of the wrinkle scales for the forehead, glabella, and other parts of facial skin were reduced 3 to 8 months after ADSC injection compared to that of the pre-treatment (Table 1, Figure 1 and Figure 2). The depth of the nasolabial groove was reduced in all eight cases, and cheek sagging was improved in addition to a fine wrinkle reduction in the lower eyelids. In a 34-year-old lady (case 1) and a 49-year-lady (case 3), an improvement in nasolabial grooves was observed 10 days and 1 month after injection (data not shown), respectively. The improvement in the lower eyelid ptosis of two male patients, cases 7 and 8, however, was confirmed 5 months and 3 months, respectively, after ADSC injection (Figure 3 and Figure 4). The mean improved ratio [1-A/B] (%)] of wrinkles and sagging in these skin areas was between 33.3% and 40%, suggesting a significant rejuvenation effect of the one-time skin injection of ADSCs (Table 1).

(2)Eyelid change after ADSC injection

In five female patients who visited our clinic one month after ADSC injection, eyelid change became clear compared to that of pre-treatment (Figure 5). Further, two of these five exhibited clearer double eyelids when they visited our clinic a few days to one week after ADSCs treatment. Since one female patient aged 58 years old did not agree to remove her false eyelashes before and after ADSCs treatment, we lost the chance to observe the effect of ADSCs on her eyelids, but her black eyeball exposure increased as shown by b/a analysis (Figure 6), suggesting an effect on her eyelids. In two male patients, case 7 showed slight upper eyelid change with a widening of the width of the eyelids, as shown by b/a analysis, and case 8 showed a slight double eyelid change visually and exhibited increased b/a (Figure 7).

(3)Reduction in facial pores via ADSC injection

In all eight cases, slight-to-remarkable pore reduction was observed clinically. Two cases (case 5 and 7) exhibited a remarkable reduction in the photographs (Figure 7). Our study indicates that significant pore reduction is expected as a result of a one-time skin injection of ADSCs.

(4)Effect of exosome fraction extracted from ADSCs on myoblast activation

Myoblasts cultured with glucose DMEM and exosome fraction produced a high level of lactic acid (Figure 8a) and reduced the level of glucose (Figure 8b) 48 h after incubation, compared to the control, which was cultured with glucose DMEM alone, suggesting a possible effect of exosome derived from ADSCs on muscle function in vivo.

## 4. Discussion

As far as we know, this is the first study to show that a single injection of cultured autosomal ADSCs into facial skin can rejuvenate aged skin, including eyelids, a few weeks to a few months after treatment. The aim of the present study is to demonstrate the rejuvenation effect of ADSC therapy on aged skin by analyzing photographs taken before and after therapy. Mesenchymal stem cells, a group of adult cells that are most abundant in bone marrow, umbilical cord blood, dental pulp, and adipose tissue, are reported to be effective at rejuvenating aged skin in pre-clinical and in vitro experimental studies, but only a few clinical studies using ADSCs are reported to show the rejuvenation effect in aged human skin. ADSCs are widely used for therapeutic purposes because of the ease of isolation and expansion in culture, but in most previous studies, stromal vascular fraction (SVF) and nanofat tissue obtained from subcutaneous fat tissue containing ADSCs were used to repair damaged tissue and rejuvenate aged skin via lipo-injection. A successful fat graft was reported, but the long-term outcome was less optimal due to the inadequate survival of the transferred fat.

In 2008, Yoshimura et al. reported that a stromal vascular fraction containing autologous adipose-derived stromal (stem) cells was relatively useful for cosmetic breast augmentation compared to lipo-injection alone [11]. SVF secretes many growth factors for fibroblasts and endothelium, and anti-inflammatory cytokines stimulate tissue regeneration by promoting the secretion of extracellular proteins and antioxidants.

Further, it is reported that in a mouse model, injected ADSCs survived for 28 days, suggesting the possible long-term active life of ADSCs in injected skin. In addition, a conditioned medium known as secretome, which contains extracellular vesicles, such as exosome, is reported to have a variety of biological activities for skin repair and rejuvenation [12]. Secretome isolated from ADSCs contains a wide range of cytokines and growth factors that are beneficial in reversing aged skin symptoms by inducing epidermal keratinocyte, dermal fibroblast activation, and proliferation [13]. The restoration of dermal thickness, wrinkle reduction, and enhanced skin texture improvement are expected after ADSCs secretome injection to the dermis. In clinical trials, MSCs were often used in combination with other procedures, such as microneedling, radiofrequency, or laser therapy, showing better improvement in macroscopic appearance, biological parameters, and histological evaluation compared to MSCs alone [14].

The present clinical study showed the apparent skin rejuvenation effect of a one-time intradermal injection of ADSCs. Further, we learned of the delayed appearance of ADSC injection effect on skin rejuvenation in one male aged 83 years, compared with young adults (data are not shown). In addition, the recovery of face sagging, even one month after injection, strongly indicates the active involvement of muscle tissue in addition to the production of collagen, elastic fibers, hyaluronic acid, and fat tissue by the injected ADSCs. Clear double eyelids and a widening of the distance between the upper and lower eyelids were observed a few days to weeks after ADSC injection, which also supports muscle activation by ADSCs. In addition, facial pore size reduction in ADSC-treated skin suggests the activation of arrector pili muscle in aged skin by ADSCs. Further, it is reported that ADSC-derived extracellular vesicles showed unique properties of signatures of mitochondrial activity and skeletal system development in addition to angiogenesis, hair growth, and dermal matrices via transcriptome approaches in comparison to dental pulp-derived extracellular vesicles [15]. Our in vitro experiments using C2C12 myoblasts showed the activation of muscle cells by the ADSC-derived exosome, supporting a possible functional involvement of muscle cells and tissue in ADSC-induced facial skin rejuvenation.

The present study demonstrates an excellent rejuvenation effect of a single intradermal injection of cultured autosomal ADSCs, and further, significant recovery of the ptotic upper eyelids impairing eye function by widening the vision field may contribute to an improved QOL of patients with ptotic eyelids. ADSC skin injection may be recommended for recovery from age-related ptosis without surgical operation of the levator aponeurosis and Muller’s muscle. Further studies are required to establish ADSC skin injection therapy as one of the most effective and efficient therapies for aged skin, by showing the best treatment conditions for total cell numbers, injection location, age, and treatment frequency. In addition, it is important to clarify the therapeutic effect of repeated injections to aged skin for patients over 80 years old with deeper shallows and highly sagged skin at the lower eyelids, nasolabial grooves, and Marionette lines which were insufficiently rejuvenated after a single skin injection.

In future studies, we aim to confirm the effectiveness of dermal injections of ADSCs on skin rejuvenation for middle- and old-aged people by treating more cases, changing the therapeutic conditions, and establishing the best treatment conditions of ADSC injections for patients with deep photoaging.

## Figures and Tables

**Figure 1 jpm-13-01162-f001:**
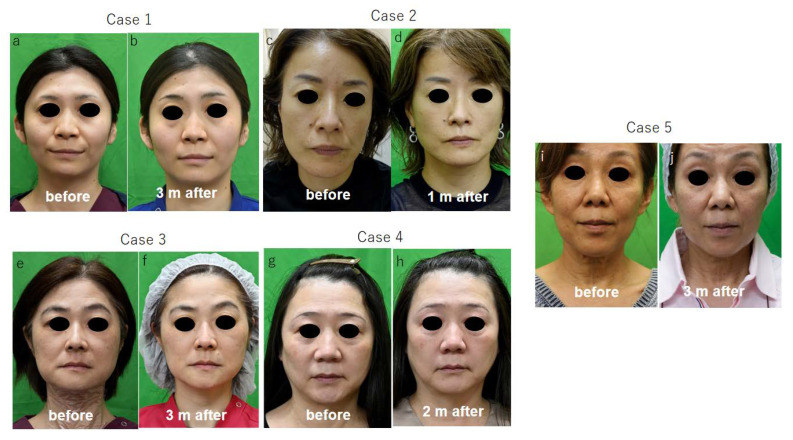
Facial skin manifestation of 5 female cases before and after treatment with intradermal injection of ADSC.

**Figure 2 jpm-13-01162-f002:**
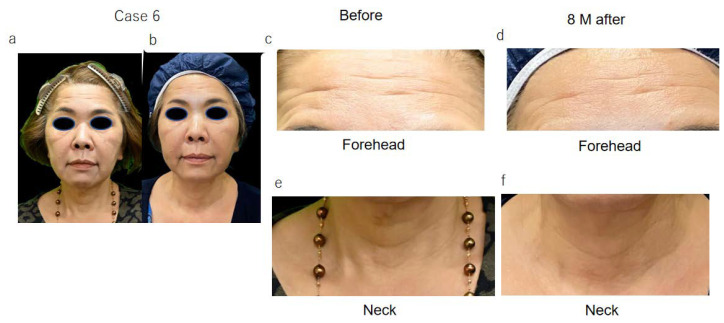
Photographs of 58 year-old lady, case 6, before (**a**,**c**,**e**) and 8 months after ADSC treatment (**b**,**d**,**f**). Reduced wrinkle of forehead 8 months after ADSC injection (**b**,**d**) and neck (**b**,**f**) and shallowing of nasolabial grooves was confirmed by comparison of (**a**) to (**b**).

**Figure 3 jpm-13-01162-f003:**
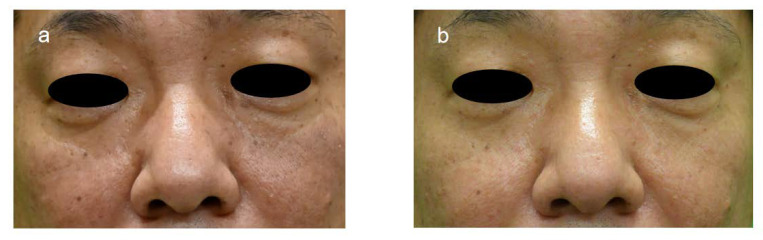
Photographs of case 7, 53-year-old male before and after ADSC treatment. Improvement of tear trough deformity and reduced hair follicle size was observed by comparing (**a**) with (**b**).

**Figure 4 jpm-13-01162-f004:**
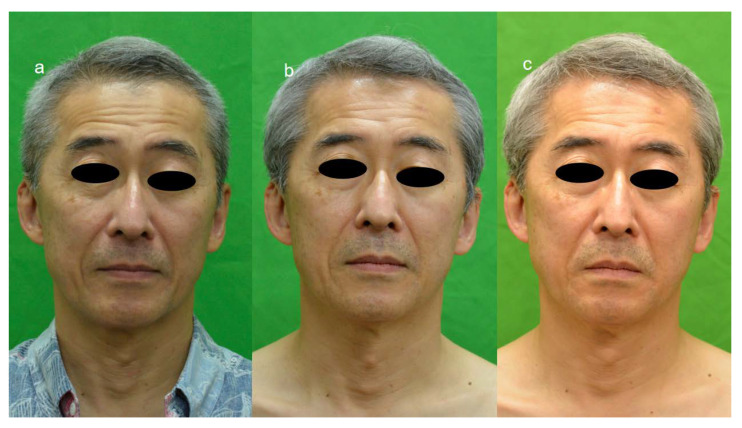
Effect of ADSCs skin injection on wrinkles and nasolabial grooves of 61-year-old male. Nasolabial grooves of (**b**) (one month after treatment) and (**c**) (3 months after treatment) were shallower than those of (**a**) before treatment, showing rejuvenation effect of ADSC skin injection of 50′s male. Tear trough deformity was improved with reduced wrinkles of lower eyelids. Wrinkles of forehead was time dependently reduced, in (**b**) (1 month after treatment) and (**c**) (3 months after treatment) V, by comparison of (**a**).

**Figure 5 jpm-13-01162-f005:**
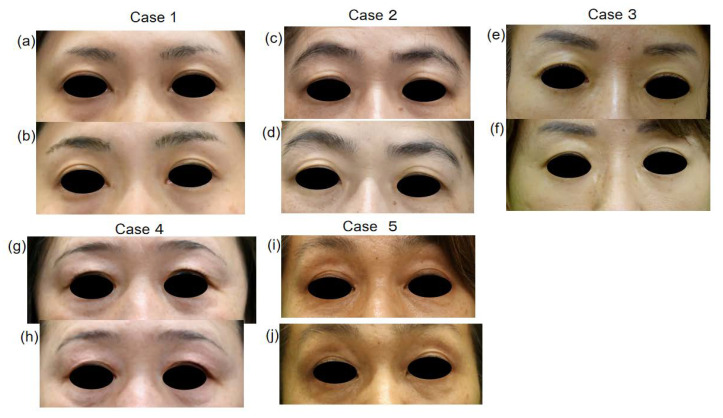
ADSC injection induced clear double eyelids. Clear eyelids of (**b**,**d**,**f**,**h**,**j**) (after treatment) compared to (**a**,**c**,**e**,**g**,**i**) (before treatment), indicate apparent effect of ADSC injection on double eyelids appearance.

**Figure 6 jpm-13-01162-f006:**
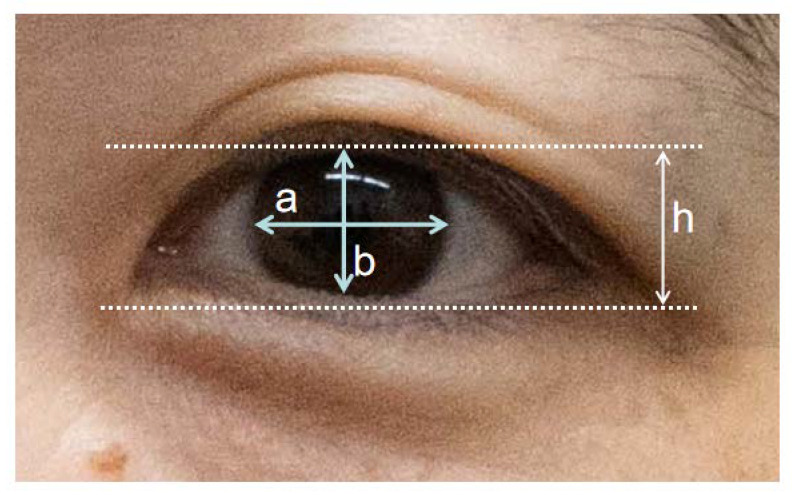
Effect of ADSC on eyelids. Eye exposure ratio (b/a) of maximal exposure length between upper and lower eyelids (b) divided by longitudinal eyeball diameter (a) was used to evaluate ADSC effect on eyelids. b/a ration increased in cases 6, 7 and 8 in whom b/a ration was evaluated, indicates that ADSC injection make double eyelids clear and makes longitudinal eyeball diameter larger.

**Figure 7 jpm-13-01162-f007:**
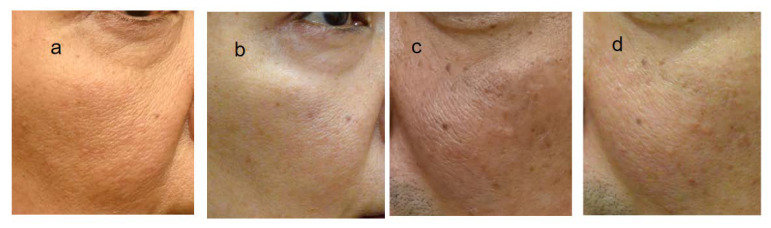
Effect on hair follicles. Hair follicles of 48y lady (**b**) and 53y male (**d**) 6 months and 5 months after ADSC treatment, respectively, are significantly smaller than those of (**a**,**c**), before treatment, showing rejuvenation effect of ADSC injection on skin texture.

**Figure 8 jpm-13-01162-f008:**
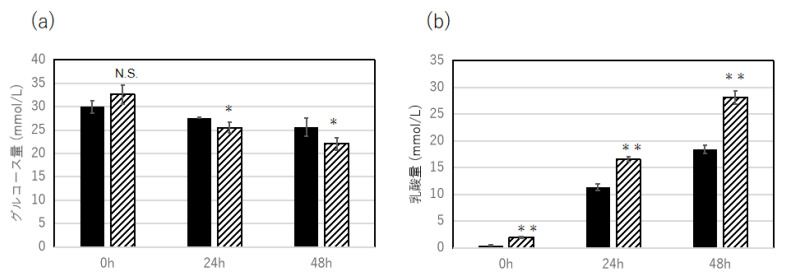
Effect of cultured ADSC-derived supernatant on C2C12 myoblasts activation. Glucose consumption rate (mmol/L) of cultured C2C12 cells slightly enhanced by the incubation with ADSC-derived supernatant for 24 h and 48 h incubation (**a**), and accumulation of glycolysis derived lactic acid level (mmol/L) was significantly increased by the incubation of ADSC-supernatant compared to untreated control (**b**). Statistical analysis was conducted using T-test. A *p*-value of less than 0.05 is considered statistically significant (**: *p* < 0.01, *: *p* < 0.05). N = 3.

**Table 1 jpm-13-01162-t001:** Dermography and clinical evaluation of cases treated with ADSCs injection.

Case No.	Age (Year)	Sex	Time of Evaluation after Treatment (Month)	Scale of Aging													
				Forehead		Glabelle		Lowereyelids		Crow’s Feet		Marionette Line		Nasolabial Grooves		Lower cheak Sagging	
				B	A	B	A	B	A	B	A	B	A	B	A	B	A
1	34	F	3	2	1	1	1	2	1	1	1	2	1	2	1	2	1
2	48	F	3	3	2	1	1	3	2	3	2	2	1	3	2	3	2
3	49	F	3	2	1	2	1	3	1	2	1	1	1	3	2	2	1
4	64	F	3	2	1	2	1	3	2	2	2	2	2	3	2	3	2
5	57	F	3	1	1	1	1	3	3	2	2	2	1	3	3	3	2
6	58	F	8	2	1	2	1	4	3	1	1	3	2	4	2	3	2
7	58	M	5	4	2	3	1	4	2	3	1	3	2	3	2	2	1
8	61	M	3	4	4	1	1	3	2	3	1	2	1	3	2	2	1
Total				20	13	13	8	25	16	17	11	17	11	24	16	20	12
Mean improved ratio [1-A/B] (%)				35		38.5		36		35.3		35.3		33.3		40	

B indicates “before treatment”, A indicates “after treatment”, F means female; Skin aging of forehead, glabelle, lowereyelid, crow’s feet, Marionette line, nasolabial grooves, lower cheak sagging was evaluated by VAS using 5 grades, 1: none, 2: slight, 3: mild, 4: moderat, 5: severe.

## Data Availability

5 members of the present study have many articles published in English which could be available via PubMed.

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
