# Peer review of "A Single Intradermal Injection of Autologous Adipose-Tissue-Derived Stem Cells Rejuvenates Aged Skin and Sharpens Double Eyelids"

_jpm, 2023, doi:10.3390/jpm13071162_

Round 1

Reviewer 1 Report

Very interesting topic, but it requires a major revision:

1. first of all, it is not framed correctly - it is a study that includes 8 patients, so it is not a case report.

2. the introduction is too long, it should have 3 paragraphs, one of which presents the objectives of the study.

3. the table must not appear in the method... the subsection related to patients must include the criteria for inclusion and exclusion from the study. All demographics are part of the results.

4. the results must be structured and evaluated objectively. It was quite difficult for me to appreciate from the photos the improvement...

5. discussions are contained in the results, not a separate chapter.

Author Response

  1. Revised figures were attached.
  2. Subjects were selected according to their aged skin alteration, such as wrinkle and sagging in face.
  3. Classification of aging was done, according to the grade of the Japanese Dermatology Association ‘s for wrinkle by the observer’s naked eye.
  4. English was edited by a US native scientist. I ask to revise the manuscript again by the English native in USA.
  5. Two recent references suggested by the reviewer were added.

Reviewer 2 Report

This research reported a single intradermal administration of ADSCs rejuvenated aged facial skin. I personally give minor revision.

1.      All the patients' mosaic pictures were viewed uncomfortably, please alter the mosaic mode.

2.      What’s the subject inclusion and exclusion criteria?

3.      Whether patients should be classified as mild to moderate to severe?

4.       Please improve English grammar mistakes.

5.      Some references are old, you can refer to the following literatures: Responsive multifunctional hydrogels emulating the chronic wounds healing cascade for skin repair (DOI: 10.1016/j.jconrel.2023.01.049), Loo HL, Goh BH, Lee L-H, Chuah LH. Application of chitosan-based nanoparticles in skin wound healing. Asian J Pharm Sci. 2022;17(3):299-332. Small extracellular vesicles secreted by urine-derived stem cells enhanced wound healing in aged mice by ameliorating cellular senescence (DOI: 10.1016/j.jmst.2020.03.014).

   Please improve English grammar mistakes.

Author Response

  1. Pictures were revised.
  2. Subjects at 30- to 80- year-old and with sagging and free of apparent skin disease in face were included, and subjects who were treated with some rejuvenation treatment ,such as botulinus toxin or hyaluronic acid injection within one year were excluded. Patients treated with our ADSC injection were asked not to take any rejuvenation treatments except everyday makeup. 
  3. Wrinkles were evaluated by naked eye according to the wrinkle criteria of the Japanese Association of Dermatology and sagging were graded written in the manuscript.
  4. English was edited by US scientist. Please see the revised manuscript.
  5. References were revised according to the reviewer comment.